



# Correcting Bias in Log-Linear Instrument Calibrations in the Context of Chemical Ionization Mass Spectrometry

Chenyang Bi[1], Jordan E. Krechmer[2], Manjula R. Canagaratna[2], Gabriel Isaacman-VanWertz[1]

[1]Department of Civil and Environmental Engineering, Virginia Tech, Blacksburg, Virginia 24060, USA
[2]Aerodyne Research Inc., Billerica, Massachusetts 01821, USA

*Correspondence to*: Gabriel Isaacman-VanWertz (ivw@vt.edu)

**Abstract.** Quantitative calibration of analytes using chemical ionization mass spectrometers (CIMS) has been hindered by the lack of commercially available standards of atmospheric oxidation products. To accurately calibrate analytes without standards, techniques have been recently developed to log-linearly correlate analyte sensitivity with instrument operating
conditions. However, there is an inherent bias when applying log-linear calibration relationships that is typically ignored. In this study, we examine the bias in a log-linear based calibration curve based on prior mathematical work. We quantify the potential bias within the context of a CIMS-relevant relationship between analyte sensitivity and instrument voltage differentials. Uncertainty in three parameters has the potential to contribute to the bias, specifically the inherent extent to which the nominal relationship can capture true sensitivity, the slope of the relationship, and the voltage differential below which
maximum sensitivity is achieved. Using a prior published case study, we estimate an average bias of 30%, with one order of magnitude for less sensitive compounds in some circumstances. A parameter-explicit solution is proposed in this work for completely removing the inherent bias generated in the log-linear calibration relationships. A simplified correction method is also suggested for cases where a comprehensive bias correction is not possible due to unknown uncertainties of calibration parameters, which is shown to eliminate the bias on average but not for each individual compound.

## 1.  Introduction

The time-of-flight chemical ionization mass spectrometer (Tof-CIMS) has been widely used for online characterization of organic compounds in the atmosphere. Gas-phase analytes are reacted with reagent ions to form analyte ions, then detected and classified by mass spectrometry. Many reagent ions have been examined, with some of the most popular being hydronium (Lindinger et al., 1998; Yuan et al., 2016), acetate (Bertram et al., 2011; Brophy and Farmer, 2016), nitrate
(Jokinen et al., 2012; Krechmer et al., 2015), $CF_3O^-$ (Crounse et al., 2006; St Clair et al., 2010), and iodide (Lee et al., 2014; Slusher, 2004). Each reagent ion accesses a different region of chemical space (Isaacman-VanWertz et al., 2017; Riva et al., 2019) and differs in its range of sensitivities, from relatively universal to highly variable. For example, proton-transfer-reaction is commonly used for measurements of less oxidized compounds with a sensitivity that varies only by a factor of up



to 3 or 4 for most analytes (Sekimoto et al., 2017). In contrast, iodide is most useful for semivolatile, oxidized compounds,
but sensitivity varies by several orders of magnitude (Iyer et al., 2016; Lopez-Hilfiker et al., 2016).

Unfortunately, these wide ranges in sensitivity pose significant issues in the quantitative measurement of ambient
atmospheres. For many atmospheric constituents, it is not possible or not feasible to calibrate using authentic standards, due
to a  lack of commercial availability and/or chemical instability (thermal lability, flammability, etc.) (Brophy, 2016). Several
approaches have consequently been developed to estimate the sensitivity of a CIMS instrument to a given analyte based on
either its physicochemical properties (e.g., dipole moment and polarizability) (Sekimoto et al., 2017) or its observed response
(e.g., induced dissociation) changing instrument conditions (Lopez-Hilfiker et al., 2016; Zaytsev et al., 2019). Estimating
instrument sensitivity based on derived relationships between sensitivity and other properties inherently carries some
uncertainty, as the relationship is unlikely to be ideal and typically includes some scatters. Nevertheless, this approach of
"derived sensitivity" is often the best (or only) tool available for calibration, so a close look is warranted into the
implications of this approach on the error of a single analyte, as well as the combined error of the sum of many analytes.

Previous work, which we discuss in detail in the following section, has examined the uncertainty in estimating a parameter
from a derived relationship (i.e., using a regression model to predict a value). Specifically, prediction of a value (e.g.,
sensitivity) from a linear model introduces no bias and has normally distributed error, but in more complex relationships
(e.g., involving log-transformations, step functions, etc.), bias and other errors may be introduced. Many of the derived
sensitivity relationships used for CIMS have more complex forms, so the overarching goal of this work is to evaluate and
correct for biases and other errors in the types of relationships used for estimating CIMS sensitivities.

We focus in this work on the calibrations of analytes in an iodide-CIMS because: (1) this measurement technique is widely
used, (2) it has orders of magnitude variance in sensitivities (Iyer et al., 2016), and (3) estimating its sensitivity often relies
on a complex (log-linear, piece-wise) sensitivity relationship. Iyer et al. (2016) have shown that the sensitivities of analytes
in an iodide-CIMS are log-linearly correlated with the binding-enthalpy of the iodide-analyte adduct, with some maximum
sensitivity that is limited by the rate of collisions between the analyte and the reagent ion. Lopez-Hilfiker et al. (2016) further
suggested that modulating voltage differences in certain component of the mass spectrometer (i.e., between the skimmer of
the small-segmented quadrupole and the entrance of the big-segmented quadrupole) can introduce de-clustering of the
iodide-molecule adduct. The parameter, $dV_{50}$, which is the voltage difference where signals of a compound are at half-
maximum, is reported to be an indicator of the binding-enthalpy of the adduct (Lopez-Hilfiker et al., 2016). Therefore, the
iodide-CIMS sensitivities can be predicted by $dV_{50}$ based on a log-linear relationship, up to a plateau of maximum sensitivity
at sufficiently high binding enthalpies.





The objective of this study is to understand the error in the calibrated mass of an analyte or the sum of multiple analytes measured by a CIMS. The work here focuses on sensitivities that are predicted using log-transformed derived relationships as in the case of the iodide-CIMS voltage scan method, but any calibration approach that relies on mathematically

transformed relationships should be studied in this manner and biases should be corrected. We first examine the problem by comparing simple linear and log-linear models used to estimate instrument sensitivity, then expand these ideas to the more complex relationship used in iodide-CIMS voltage scanning, and finally provide and evaluate corrections to reduce or even remove the bias.

## 2. Prior work on uncertainty analysis

### 2.1 Linear fits

In some cases, the sensitivity of an instrument can be estimated from a direct linear fit to a property or parameter. For example, sensitivity of a flame ionization detector is linearly correlated with oxygen-to-carbon content of an analyte (Hurley et al., 2020). In a linear model such as this, the average residual of the fit (i.e., the difference between the true sensitivity and the predicted sensitivity) will necessarily be equal to zero. In other words, there is no difference between average true and

average modeled sensitivity. The sensitivity of any given analyte might be uncertain, but those uncertainties are normally distributed around the model, so the potential overprediction is equal in scale to the potential underprediction. The average sensitivity measured for each analyte is therefore unbiased, and the summed mass of multiple ions is consequently unbiased. Specifically, relative uncertainty, $\sigma_{sum}$, in the summed mass or concentration, $C_{sum}$, of N analytes is the sum of the squares of the relative uncertainty in each individual analyte, $\sigma_i$ and their individual concentrations, $C_i$:

$$\sigma_{sum} = \frac{\sqrt{\sum_i^N C_i^2 \sigma_i^2}}{C_{sum}} \tag{1}$$

In cases where relative uncertainty of each analyte is equal (e.g., "instrument uncertainty is 20%"), $\sigma_1 = \sigma_2 = \cdots = \sigma_N$ and Equation 1 can be re-written as:

$$\sigma_{sum} = \frac{\sqrt{\sum_i^N C_i^2}}{C_{sum}} \sigma_i \tag{2}$$

This equation has two extreme conditions. When one compound dominates total mass, $C_{sum}$ is essentially equal to $C_i$ and this

equation collapses to:

$$\sigma_{sum} = \frac{\sqrt{C_i^2}}{C_i} \sigma_i = \sigma_i \tag{3}$$

In this case, relative uncertainty is equal to that of a single analyte. At the other extreme condition, when all *N* analytes are equal in concentration, this equation collapses to:





$$\sigma_{sum} = \frac{\sqrt{NC_i{}^2}}{NC_i} \sigma_i = \frac{\sigma_i}{\sqrt{N}} \tag{4}$$

In most cases, neither a single analyte will dominate summed mass, nor will all analytes be evenly distributed, so uncertainties in the summed mass of real-world measurements likely fall between the extremes of $\sigma_i/\sqrt{N}$ and $\sigma_i$. From Equations 3 and 4 it is clear then that, when calibration relies on linear fits, relative uncertainty in summed mass is generally lower than uncertainty in the mass of an individual analyte. In other words, as a set of analytes gets larger, their average predicted sensitivities is increasingly well described by the average model.

**2.2 Log-transformed fits**

It is tempting to assume that the conclusions drawn from linear fits are generalizable: that the sum of many analytes is less uncertain than any given analyte. However, this conclusion has some truth, as well as some limitations, when a mathematical transformation (e.g., the logarithm) is applied to data to linearize it. The case we address here is specifically when log(sensitivity), not sensitivity, is correlated to some other parameter, as in the case of iodide-CIMS voltage scanning. What

we present here is substantively similar to the treatment by Miller (1984) of the case of linear fits to natural-log-transformed data, through the lens of its implications for atmospheric measurements.

A linear fit through log-transformed data can be described as:

$$\log(Y) = \alpha + \beta X + \varepsilon \tag{5}$$

where the log-transformed value of Y is described by two coefficients ($\alpha$ and $\beta$) describing a linear relationship with X, and an error term, $\varepsilon$, describing deviation in the true value from the fit. In such a fit, the error term will be evenly distributed in logarithmic terms, meaning it is log-normally distributed in linear terms (i.e., $\varepsilon$ is normally distributed).





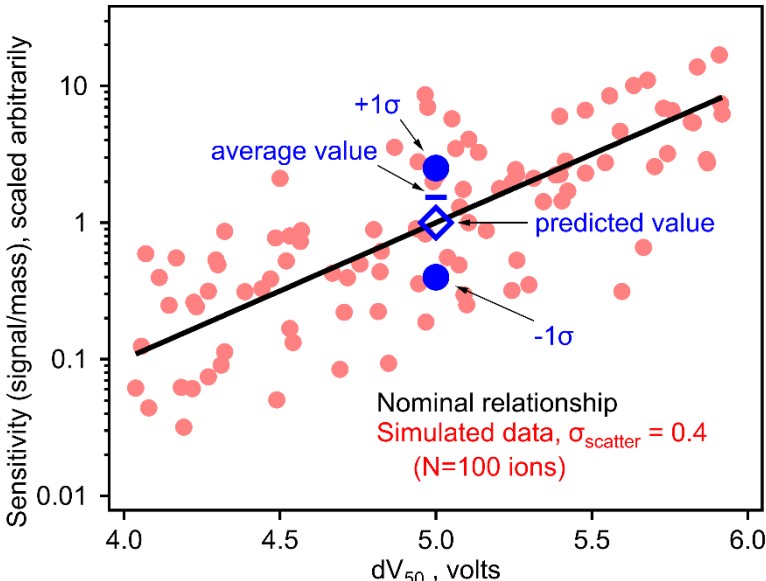

**Figure 1.** Simulated sample relationship between log(sensitivity) and $dV_{50}$ for an iodide-CIMS. Nominal relationship is the black line, with simulated sensitivities for 100 analytes around this relationship as circles. Blue markers demonstrate bias in the average value as described in the text.

To understand the effect of this error term on a real instrument, we consider a thought experiment presented in Figure 1, though the discussion here applies to linear fits through any log-transformed data. A distribution of points is shown with a normal distribution of "scatter" around an average linear fit describing the relationship between log(sensitivity) and the parameter, $dV_{50}$, that is an empirical description of analyte-reagent ion binding-enthalpy. Sensitivities of 100 simulated analytes are shown with a distribution of scatter described by $\sigma_{scatter} = 0.4$ log units (i.e., a factor of 2.5, similar to previously-estimated uncertainty in an iodide-CIMS (Isaacman-Vanwertz et al., 2018)). Consider two analytes of $dV_{50} = 5.0$ V (i.e., blue circles in Figure 1), which have an equal probability of occurring at one sigma above or below this fit. Using this log-linear fit, the sensitivity that would be assigned to both analytes is $10^0 = 1$ (in units of signal per mass, scaled arbitrarily). However, one analyte has a true sensitivity of $10^{0.4} = 2.5$ signal/mass while the other has a true sensitivity of $10^{-0.4} = 0.4$ signal/mass. The average sensitivity of these two components is therefore 1.45 signal/mass, 45% higher than the predicted value. In other words, uncertainty in log terms is implicitly "factor"-based uncertainty as opposed to "percentage"-based uncertainty, and a factor of 2.5 times (i.e., 0.4 log unit) larger is a higher difference than a factor of 2.5 times smaller. Taking this example one step further, consider an environment in which both analytes are present in equal mass, e.g., one mass unit each = two mass units total. Signal generated by this instrument from both analytes would equal 0.4 + 2.5 signal units = 2.9 signal units. In turn, the 2.9 signal units would be interpreted using the predicted sensitivities of 1 signal/mass, calculating a total mass of 2.9 mass units, 45% higher than the true mass of 2 mass units. Summing increasingly large numbers of ions does not remove this bias. Instead, a correction must be made to the log-linear model to account for this difference between true and predicted average sensitivity.






Correcting this bias requires a proper consideration of the true average of the error term, ε, in linear terms. The true value of Y can be calculated as:

$$Y = 10^{\alpha}10^{\beta X}10^{\varepsilon} \tag{6}$$

The median of a log-normal distribution is equal to the median of the log-transformed distribution, so the median value of Y

is correctly represented by this equation. However, as observed in the example shown in Figure 1, the mean value of a log-normal distribution is higher than the mean of the log-transformed distribution. Specifically, the mean value of a log-normal distribution with a width of σ and a median of 0, for any logarithm base, B is:

$$\text{Mean of a base-B-log-normal distribution} = e^{\frac{1}{2}(\ln(B)\sigma_{scatter})^2} = B^{\ln(B)\frac{1}{2}\sigma_{scatter}^2} \tag{7}$$

Both forms of this equation given are equivalent, and for a natural-log-normal distribution, it collapses to the expected form

of $e^{\frac{1}{2}\sigma_{scatter}^2}$ (Miller, 1984). Equations 6 and 7 can be combined to yield a full equation for accurately estimating the mean expected value of Y using a linear fit through base-10-log-transformed data:

$$\hat{Y} = 10^{\alpha}10^{\beta X}10^{\ln(10)\frac{1}{2}\sigma_{scatter}^2} \tag{8}$$

Implementation of this error-correction term removes the expected bias. Figure 2 demonstrates the bias in average mass in the simple scenario discussed above as a function of the scatter around a log-linear model. Inherent bias in the sensitivity,

and thus measured mass, of ions is reasonable for small values of $\sigma_{scatter}$, but quickly becomes substantial with increasing $\sigma_{scatter}$. Introducing the bias correction term removes bias entirely. The magnitude of this bias is independent of assumptions about the relationship shown in Figure 1, such as its slope or the range of $dV_{50}$ across which is applied.





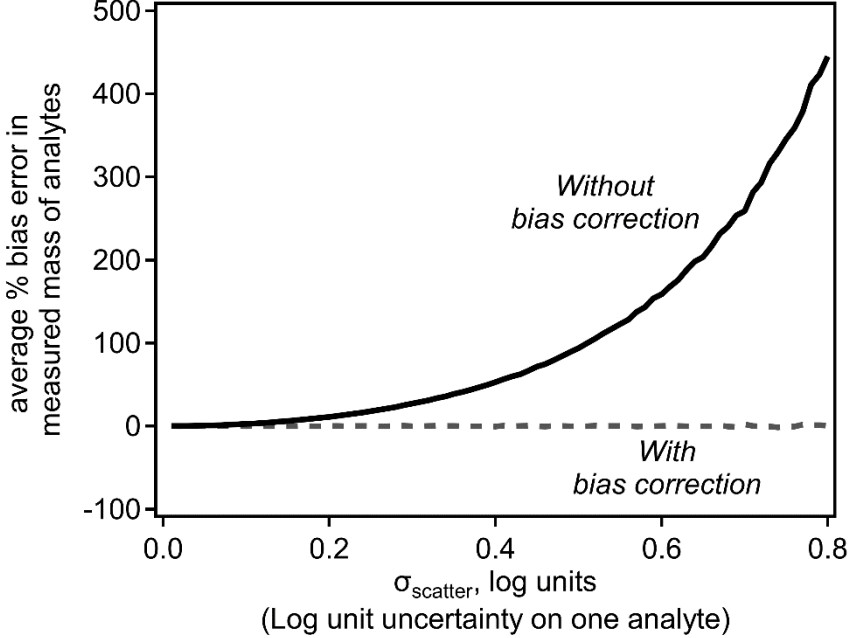

**Figure 2.** Average of bias (percentage) in mass of one or multiple analytes with and without bias correction.

## 3. Method for quantifying error in sums of analytes


Equations 4 and 8 suggest two important conclusions: (1) summing multiple analytes reduces the uncertainty in the summed concentration, and (2) analytes and sums of analytes calibrated using log-transformed relationships are inherently biased. To explore the combined effects of these two conclusions, we perform a Monte Carlo analysis that simulates the real-world application of log-transformed sensitivity models. N number of simulated analytes are generated with a randomly assigned

"true sensitivity" defined by the relationship shown in Figure 1 with a Gaussian distribution of error, $\sigma$. Each analyte is assigned a random "true sampled mass" spanning six orders of magnitude (i.e., $10^{-3}$ to $10^{3}$ arbitrary mass units). A simulated signal produced by each analyte is calculated by multiplying its true sensitivity by its true sampled mass. The nominal log-linear model is used to estimate a fitted sensitivity for each analyte, which is used to convert the signal to the fitted mass of an analyte. The summed fitted mass of all N analytes is compared to the summed true sampled mass to calculate the error in

the fitted mass; 100,000 such simulations of N analytes yield a probability distribution of expected error.

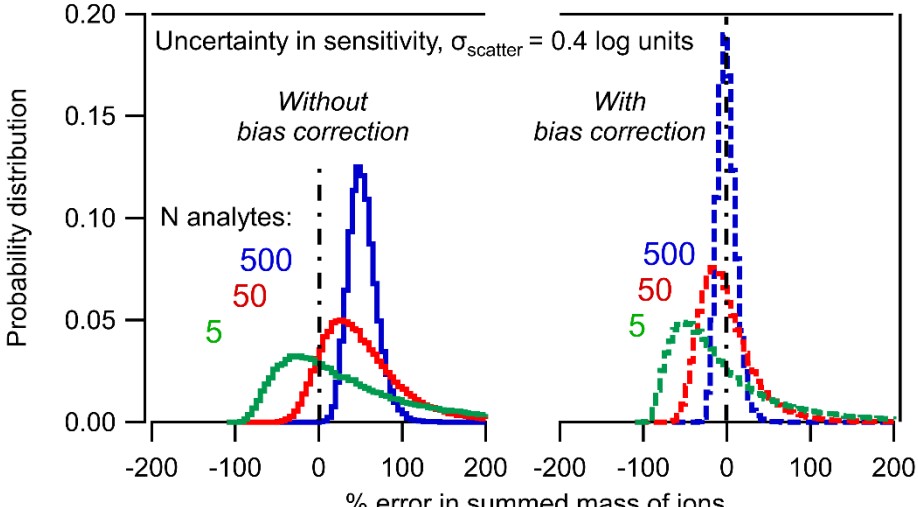

**Figure 3.** Probability distribution of percent error in the summed mass of N analytes based on the Monte Carlo analysis described in the text. Distributions of 5, 50, and 500 analytes are shown (left) without bias correction and (right) with bias correction using Equation 8.

The combined effects of the two statistical trends implied by Equations 4 and 8 are clear in Figure 3. As the number of analytes, N, increases, the sum of the mass converges toward a tighter distribution of uncertainty (i.e., the sum becomes less uncertain). However, the mass to which the distribution converges is inherently biased. In other words, the sum of 5 analytes may span a wide range of potential error, but on average they will be biased high by ~50%. Increasing the number of analytes just improves the probability that the sum is ~50% too high. The sum of 500 analytes, each with an uncertainty of a

factor of 2.5 has high precision, but inherently biased accuracy.

        This approach assumes that true sensitivities are not perfectly represented by the nominal relationship (i.e., scatter is "real"); this is in contrast to the case in which each analyte is actually truly described by the fit and deviations are due to measurement uncertainties (i.e., scatter is measurement error). If the latter is discovered in subsequent literature to be the

case, no bias would truly exist and the work in this manuscript is extraneous. However, we believe the more likely case is that the scatter is a real consequence of the calibration approach for two reasons. Firstly, it is unlikely that an empirically derived relationship captures with perfect fidelity the sensitivity of an analyte. Secondly, because an iodide-CIMS classifies analytes by elemental formula with no regard to molecular structure, the $dV_{50}$ of each analyte (i.e., ion) is typically some combination of multiple compounds (Bi et al., 2020). It therefore inherently represents some composite of a distribution of

analytes and is unlikely to equally represent all analytes in the mixture. Nevertheless, scatter measured by a real instrument provides some insight into true scatter; imperfect measurements of many compounds scattered around the nominal relationship would yield the nominal relationship with some uncertainty that represents the true variability (at least to some





greater or lesser degree). For the purposes of real-world instruments, then, we suggest that it is reasonable to use observed uncertainty in model parameters as an estimate of their true variability and will do so throughout this work.

**4.    Expanding to CIMS-specific parameters**

**4.1 Sources of uncertainty**

So far, this work has treated a fairly simple case: normally distributed error in log-transformed data. However, in the case of an iodide-CIMS calibration using voltage scans, this may not accurately represent the form of uncertainty. The true relationship between log(sensitivity) and $dV_{50}$ has several parameters, each of which could be uncertain or may represent a

central tendency of an inherently imperfect relationship. Figure 4 shows the nominal relationship between sensitivity and $dV_{50}$ of a form that is typically considered for an iodide-CIMS using voltage scans, as well as an illustrative potential spread of true sensitivities around this relationship. At some $dV_{50,max}$, the instrument reaches maximum sensitivity, $S_{max}$, and it might be reasonable to expect that analytes closer to this value adhere more closely to the general relationship than compounds that diverge significantly from maximum sensitivity. In this case, variability in sensitivity may itself be partly

(but perhaps only partly) a function of $dV_{50}$ (i.e., heteroscedastic). Note that while compounds near maximum sensitivity are generally well predicted, the nominal relationship in Figure 4 assumes that sensitivities of low-sensitivity analytes may diverge by roughly an order of magnitude from the general trend.

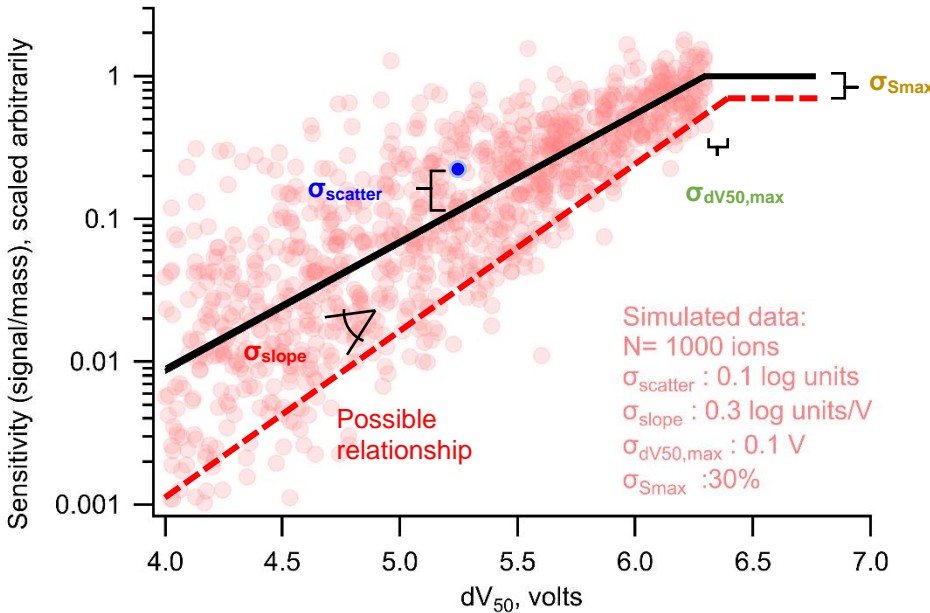

**Figure 4.** Nominal (black line) log-linear relationship between sensitivity and $dV_{50}$ following a typical iodide-CIMS calibration form, with labels on the four sources of uncertainty described in the text. Each circle represents one ion with the

distribution of uncertainties listed. Dashed line illustrates one possible relationship representing one standard deviation away from each nominal value.





The relationship shown in Figure 4 is defined by four critical parameters that may have some uncertainty or may deviate from the nominal relationship. The distribution of sensitivities can be described by some distribution in each of the four parameters, in the units and forms they have been previously considered (Isaacman-Vanwertz et al., 2018).

1) $\sigma_{scatter}$: Scatter in true sensitivity around the nominal relationship (i.e., the extent to which the average relationship inherently describes the data). Units of log units of sensitivity.

2) $\sigma_{slope}$: Variability in the slope of the relationship between log(sensitivity) and $dV_{50}$. Units of log units of sensitivity per volt.

3) $\sigma_{dV50,max}$: Variability in the inflection point, the $dV_{50}$ voltage at which sensitivity reaches its maximum. Units of volts.

4) $\sigma_{Smax}$: Extent to which the nominal maximum sensitivity describes the sensitivity of compounds that are expected to be maximally sensitive. Units of percent.

Each deviation from the nominal relationship will lead to inherent bias as in the simple log-transformed example discussed above. The exception to this issue is the fourth source of variability, variability in maximum sensitivity. Because this parameter is typically known reasonably well, uncertainty is low and best considered as a percentage. Uncertainty in this parameter is therefore not in log terms, and does not introduce bias (i.e., 10% lower and 10% higher are equally different from the nominal maximum sensitivity).

## 4.2 Bias correction

To correct for the three potential sources of bias, we introduce Equation 9 to calculate the expected sensitivity, S, of an analyte of a given $dV_{50}$:

$$\hat{S} = S_{max}(10^{slope \times \Delta dV_{50}})\left(10^{\ln(10)\frac{1}{2}\sigma_{scatter}^2}\right)\left(10^{\ln(10)\frac{1}{2}(\Delta dV_{50} \times \sigma_{slope})^2}\right)\left(10^{\ln(10)\frac{1}{2}(slope \times \sigma_{dV50,max})^2}\right) \tag{9}$$

A critical term in this calculation is the extent to which the $dV_{50}$ of an analyte is below the nominal inflection point $dV_{50,max}$, which is defined as going to 0 in the region of maximum sensitivity, $\Delta dV_{50} = \max(dV_{50,max} - dV_{50}, 0)$. The slope is defined as change in log(sensitivity) per unit $\Delta dV_{50}$, and is therefore necessarily a negative value (i.e., sensitivity decreases with $\Delta dV_{50}$). The first two terms in this equation (i.e., $S_{max}(10^{slope \times \Delta dV_{50}})$) constituent the nominal relationship, while the last three terms introduce corrections for bias due to $\sigma_{scatter}$, $\sigma_{slope}$, and $\sigma_{dV50,max}$, respectively. We note that the first two terms are identical in form to the sensitivity equation used in previous work (Isaacman-Vanwertz et al., 2018), except Equation 9 excludes an additional correction term ("$S_0$") that is outside the scope of the present work but is typically included to account for partial declustering at $\Delta dV_{50} = 0$.





Unlike in the simple case of $\sigma_{scatter}$, note that bias correction factors for $\sigma_{slope}$ and $\sigma_{dV50,max}$ are not independent of parameters in the nominal relationship. Bias caused by $\sigma_{slope}$ increases with the range of $dV_{50}$ across which the relationship is applied. Bias caused by $\sigma_{dV50,max}$ increases with the slope, which makes sense when considered at its extreme - if there were no decrease in sensitivity with $dV_{50}$ then the inflection point is irrelevant. Given these dependencies, the scope of bias and the

efficacy of the bias correction term must be explored using some approximation of typical CIMS conditions. We use for this work the calibration parameters used by Isaacman-Vanwertz et al. (2018): $dV_{50,max}$ = 6.3 V, slope = -0.9 log units sensitivity per volt, up to a maximum $\Delta dV_{50}$ = 2.3 (a minimum effective sensitivity was applied by Isaacman-Vanwertz et al. (2018), which is irrelevant to this work). Using these bounding conditions, the bias introduced by the model parameters is shown in Figure 5, in which the other sources of variability are held at 0 to isolate the effect of each parameter. As in Figure 2, bias

quickly increases with $\sigma$ for all parameters except $\sigma_{Smax}$ (as expected). The correction factors introduced in Equation 9 almost fully remove all bias.

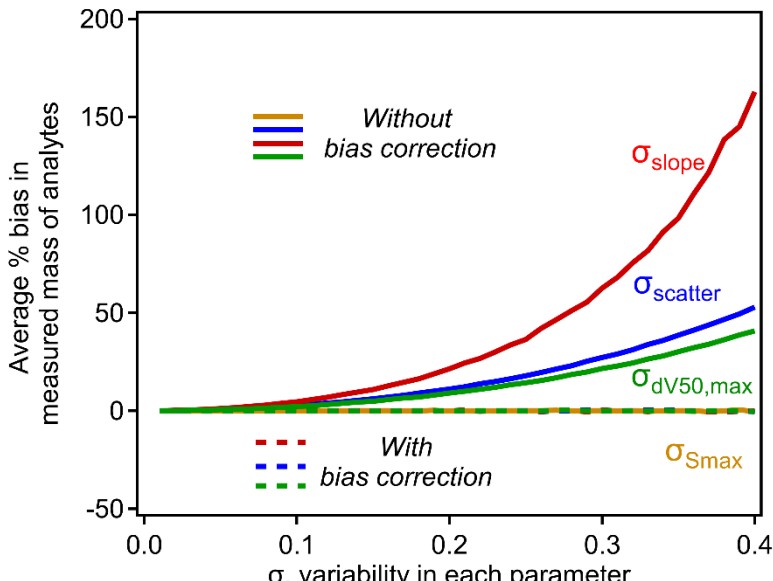

**Figure 5.** The influence of $\sigma_{scatter}$, $\sigma_{slope}$, $\sigma_{dV50,max}$, and $\sigma_{Smax}$ on the average percent bias in analytes calibrated using the typical iodide-CIMS relationship shown in Figure 4. Each curve is calculated empirically using a simulation of N = $10^6$ analytes, with operating conditions of $dV_{50,max}$ = 6.3 V, slope = -0.9 log units sensitivity per volt, up to a maximum $\Delta dV_{50}$ =
2.3, following Isaacman-VanWertz et al. (2018). Uncertainty in each parameter not being investigated is held at 0.

To examine the combined effect of variability in all four parameters, we investigate the conditions described for a real-world iodide-CIMS by Isaacman-Vanwertz et al. (2018). Uncertainty was estimated based on reported values in that work: $\sigma_{scatter}$ = 0.2 log units, $\sigma_{slope}$ = 0.125 log units per volt, $\sigma_{dV50,max}$ = 0.125 volts, $\sigma_{Smax}$ = 85% (calculated as the approximate standard deviation of their two reported possible values for $S_{max}$). No value for $\sigma_{scatter}$ was actually reported as no measurements were
available to constrain the inherent scatter in sensitivity, so 0.2 log units is assigned here as an estimate that produces approximately the same average uncertainty reported in that work for individual ions (a factor of ~2.5).



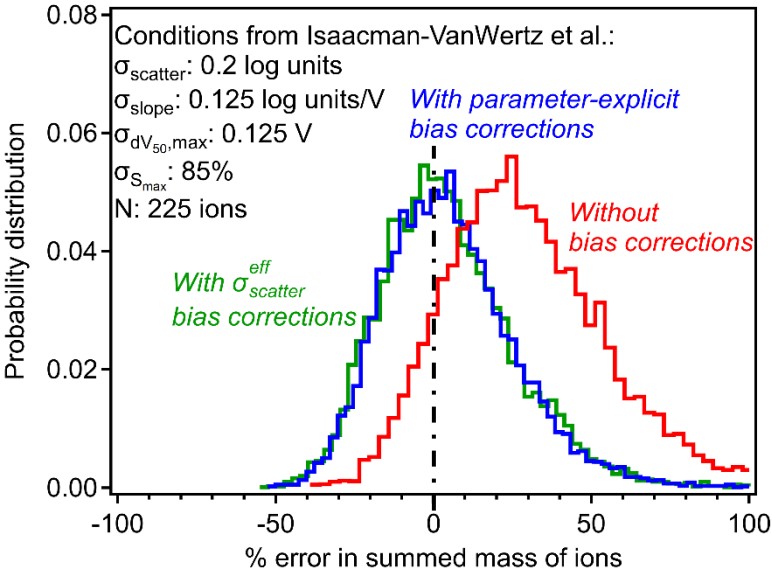

**Figure 6.** Errors in summed mass of ions with and without bias corrections in the case study of the CIMS conditions described by Isaacman-VanWertz et al. (2018), represented by the listed uncertainties and number of analytes. Probability distributions of errors are shown for calibration without including any bias correction (red), including the parameter-explicit bias correction described by Equation 9 (blue), and including the simplified bias correction described by Equations. 10 and 11 (green).

The example shown in Figure 6 provides a case study to examine the importance and the limitations of bias correction. Without introducing the correction parameters, the sum of 225 analytes (ions measured by iodide-CIMS) is expected to yield a mass roughly 30% too high, with a range of possible measurements spanning from negative error to nearly a factor of 2. As described in Section 2.2, this 30% bias in the sum is caused by an average 30% in each individual analyte, so the bias exists for one analyte as well as for the sum of analytes. Introducing the correction parameters in Equation 9 removes this bias and tightens the distribution, but the range of possible sums is still substantial. In the work upon which this case study is based, Isaacman-VanWertz et al. (2018) used a similar Monte Carlo approach to calibrate all ions, explicitly considering a distribution of uncertainty in calibration parameters, so likely avoided introducing bias. Notably, they estimated that a factor of three uncertainty in any given analyte led to an uncertainty of ~60% in the sum of the 225 ions measured, comparable to the width of the distribution shown in Figure 6. The approach described here alleviates the need to perform a full Monte Carlo approach in future work seeking to calibrate large numbers of ions, instead using Equation 9 to remove the bias in average sensitivities.

## 5. Corrections in real-world applications





To remove bias in CIMS calibration, the correction terms in Equation 9 should be included in the calculation of an analyte's sensitivity. However, in many real-world cases, the number of calibrants to establish the log-linear relationship is limited (e.g., fewer than 10 in Lopez-Hilfiker et al. (2016) and Mattila et al. (2020)), so it may not be feasible to separately treat uncertainty in all four parameters. A simplification of the detailed, parameter-explicit approach here could instead treat all forms of uncertainty as some average residual between the nominal and true sensitivities with an effective scatter, $\sigma_{scatter}^{eff}$. Such an approach would apply only the first correction term using this average scatter, ignoring the terms dependent on $dV_{50}$ and slope, and implicitly assumes that uncertainty is homoscedastic. This simplified approach, shown below in Equation 10, is mathematically equivalent to the basic log-transformed case of Equation 8 and roughly works for low to moderate $\sigma_{scatter}^{eff}$, but loses skill as the slope and the range in $dV_{50}$ increase.

$$\hat{S} = S_{max}(10^{slope \times \Delta dV_{50}}) \left( 10^{\ln(10)\frac{1}{2}\sigma_{scatter}^{eff^2}} \right) \tag{10}$$

Application of Equation 10 represents a more feasible approach to the implementation of bias correction under many real-world scenarios than the full, parameter-explicit form of Equation 9, but requires a careful consideration of the best approach to estimate $\sigma_{scatter}^{eff}$. In the specific case that $\sigma_{scatter}$ is the only source of uncertainty (i.e., $\sigma_{slope} = \sigma_{dV50,max} = 0$ ), $\sigma_{scatter}^{eff}$ must equal $\sigma_{scatter}$ and error is homoscedastic. Because $\sigma_{scatter}$ is by definition a description of the error in the model relationship, it can be estimated as the standard deviation of the residual of the log-linear fit ($\sigma_{residual}$) and this value must represent a reasonable estimate of $\sigma_{scatter}^{eff}$. However, a non-negligible caveat to this approach is that $\sigma_{Smax}$ quantiatively impacts the residual of the log-linear fit, but does not introduce bias and thus should not influence the bias correction term. Consequently, the effect of $\sigma_{Smax}$ needs to be removed from the residual before using it as an estimate of $\sigma_{scatter}^{eff}$. Fortunately, uncertainty in $S_{max}$ is often reasonably well constrained based on experimental parameters (e.g., uncertainty in the calibration of a maximum senstivity analyte), so $\sigma_{scatter}^{eff}$ can be estimated as:

$$\sigma_{scatter}^{eff} = \sqrt{\sigma_{residual}^2 - \sigma_{Smax(log)}^2} \tag{11}$$

where $\sigma_{Smax(log)}$ is the log-equivalent uncertainty in $S_{max}$, which is typically considered in linear terms. For example in the case of $\sigma_{Smax} = 10\%$ (i.e., a factor of 0.9), the log-equivalent uncertainty $\sigma_{Smax(log)}$ is, $\log(0.9) \times (-1) = 0.045$. This linear-to-log conversion is only meaningful for relatively low uncertainty ($< \sim 50\%$), for which $\sigma_{Smax(log)}$ can be estimated as:

$$\sigma_{Smax(log)} = -\log(1 - \sigma_{Smax}) \tag{12}$$

For uncertainty in $\sigma_{Smax}$ beyond 50%, the conversion is provided as Equation S1, but uncertainty is probably sufficiently high that it should be considered in log terms in any case. In some cases, $\sigma_{Smax}$ may not be available, so in the Supporting Information, we examine alternative statistical parameters as $\sigma_{scatter}^{eff}$ but find that Equation 11 is most effective in eliminating the bias.




As shown in Figure 6 (green line), the average bias in the calibrated mass of analytes can be fully eliminated using the simplified $\sigma_{scatter}^{eff}$ bias corrections. However, a significant shortcoming of this simplified approach is its implicit assumption of homoscedastic error. Some single average correction will necessarily underestimate the bias in some analytes and overestimate the bias in others. Because uncertainty is expected to increase with decreasing sensitivity, this simplified

correction will lead to a systematic bias toward overcorrecting high sensitivity analytes and undercorrecting low sensitivity analytes. This issue is demonstrated in Figure 7, in which the simplified bias correction (green line) represents some average representation of the true bias correction (blue). The effect of this issue is strongly dependent on the relative importance of each source of error. Limitations of the simplified approach are more severe in cases where heteroscedastic errors (e.g., $\sigma_{slope}$) are significant (Figure S5). Not enough data is yet available in the literature to determine the relative importance of

uncertainties in each parameter, so the potential downsides of the simplified approach are not yet well constrained. Therefore, parameter-explicit bias correction should be implemented in cases where all four parameters can be reasonably estimated, but a simplified approach remains reasonable.

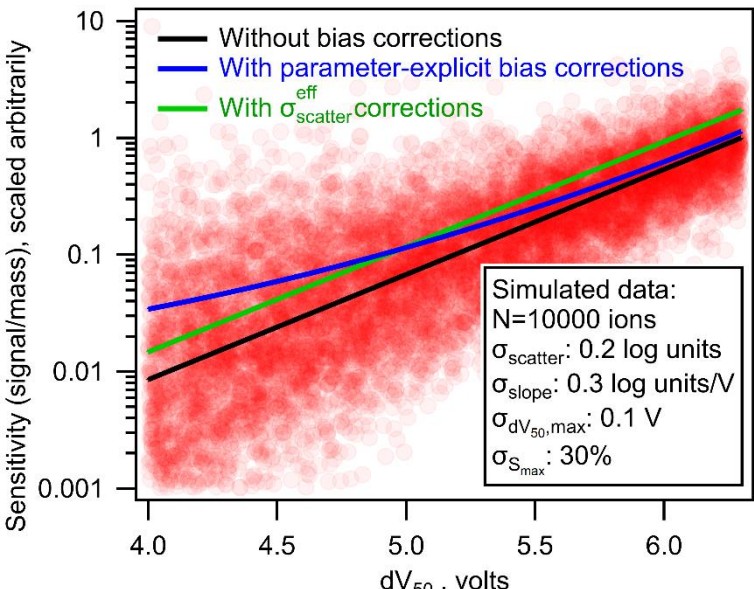

**Figure 7.** The nominal (black line), parameter-explicit bias corrected (blue line), and $\sigma_{scatter}^{eff}$ bias corrected (green line) log-linear relationship between sensitivity and $dV_{50}$. Each pink circle represents one ion with uncertainties listed.

An additional value of $\sigma_{scatter}^{eff}$ is that it can be considered as an indicator of magnitude of the potential bias in retrospective analyses of past datasets. In two previous studies implementing the voltage scan calibration, we found that the calculated $\sigma_{scatter}^{eff}$ can be as low as 0.012 (Mattila et al., 2020), or as high as 0.29 (Iyer et al., 2016; Lopez-Hilfiker et al., 2016).

However, these two extremes represent voltage scanning of two different instrument voltage regions and are calculated using



limited number of calibrants. The potential range of the $\sigma_{scatter}^{eff}$ therefore remains unclear and future work is needed to examine this approach in real-world applications, and the real potential for bias in voltage scanning approaches. Furthermore, the number of calibrants in a voltage scan calibration is often limited due to the lack of commercially available standards covering the entire sensitivity range, so $\sigma_{residual}$ (and thus $\sigma_{scatter}^{eff}$) may not adequately capture the true scatter of

residuals.

## 6.    Conclusions

In this work, we examine uncertainty in the case where instrument sensitivity is itself a function of some parameter, with a focus on uncertainty in the summation of multiple analytes. We show that when sensitivity is a linear function of a parameter, the sum of multiple analytes necessarily has lower relative uncertainty than any given analyte. However, when a

log transformation is used to linearize the relationship between sensitivity and a parameter, as in the case of an iodide-CIMS, an inherent bias is introduced into the sensitivity of analytes. While summing multiple analytes increases the precision of the sum, the bias can only be eliminated by specifically introducing correction terms to the relationship. Although the discussions of this work mainly focus on iodide-CIMS, we believe that this correction can be applied to other CIMS or more broadly, other atmospheric measurement instruments using log-linear calibration relationships.


The correction terms introduced in this work for both the general case of log-transformed relationships, and the special case of an iodide-CIMS (i.e., Equation 9), fully remove this bias. We propose that these corrections terms should be introduced into any such calibration schemes in future work in order to minimize bias and reduce uncertainty in the literature. Given that real-world calibration scenarios are complex and consequently not all parameters have known uncertainties, we suggest that,

at least, a term to correct for the average observed scatter around the nominal relationship, i.e., Equation 10, should be incorporated in calibrations to remove a major portion of the bias. For the convenience of method users, we summarize correction procedures as a step-by-step guidance to apply the simplified bias correction method in Table 1.





**Table 1.** Step-by-step guidance to apply the simplified bias correction method

| | |
|---|---|
| *Step 1* : | Fit the log-transformed sensitivities with obtained $dV_{50}$ using a linear relationship. |
| *Step 2* : | Calculate $\sigma_{residual}$, the standard deviation of the residuals of the fit, $\log(S_{measured,i}) - \log(S_{fitted, i})$. |
| *Step 3*: | Calculate $\sigma_{Smax(log)}$[*]: |

$$\sigma_{Smax(log)} = -\log(1 - \sigma_{Smax})$$

| | |
|---|---|
| *Step 4* : | Calculate $\sigma_{scatter}^{eff}$ using obtained $\sigma_{residual}$ and $\sigma_{Smax(log)}$ |

$$\sigma_{scatter}^{eff} = \sqrt{\sigma_{residual}^2 - \sigma_{Smax(log)}^2}$$

| | |
|---|---|
| *Step 5* : | Calculate bias-corrected sensitivity by adding a correction term, $10^{\ln(10)\frac{1}{2}\sigma_{scatter}^{eff}{}^2}$, to the nominal fit: |

$$\hat{S} = S_{max}(10^{slope \times \Delta dV_{50}})\left(10^{\ln(10)\frac{1}{2}\sigma_{scatter}^{eff}{}^2}\right)$$

[*]: Equation S1 should be used to calculate $\sigma_{Smax(log)}$ when $\sigma_{Smax} > 50\%$.


However, we do recommend that this simplified approach be used cautiously to avoid overcorrections of sensitivity for more sensitive analytes and undercorrections for less sensitive ones. While data is limited on the uncertainty in each calibration parameter and the relative merits of simplified vs. parameter-explicit correction, bias-corrected results are expected to be more accurate than uncorrected values and some form of bias correction should be introduced into instrument calibrations

relying on log-transformed calibrations.

**Data availability**

All raw and processed data collected as part of this project are available upon request.

**Author contributions**

GIVW developed the thought experiment. CB led the consequent data simulation and analysis under the guidance of GIVW.
JEK and MRC contributed to the development of the theory of the described approach. CB and GIVW prepared the manuscript with contributions from all other authors.

**Competing interests**

JEK and MRC are employed by Aerodyne Research, Inc., which commercializes CIMS instruments for geoscience research.

**Acknowledgments**



This work is primarily supported by the Alfred P. Sloan Foundation Chemistry of the Indoor Environment Program (P-2018-11129).

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
