# Peer review of "Correcting Bias in Log-Linear Instrument Calibrations in the Context of Chemical Ionization Mass Spectrometry"

_Atmospheric Measurement Techniques, 2021_

## Author Comment (AC1)

The authors would like to thank both reviewers for the feedback on the manuscript. Those comments are insightful, and the suggestions have helped improve the quality of the manuscript substantially. We have made revisions to the manuscript according to the reviewers' comments and the extra experimental findings. The colorings of text in the reviewer response are:

- Light blue: Original reviewer comments
- Dark blue: Text added in the revision while strikethrough words are the text deleted in the revised manuscript.
- Black: Original text in the submitted version of the manuscript and authors' response to the comments and others.

Note that the line number in the response is based on the revised clean-version manuscript.

**Reviewer 1:**

This manuscript presents a method for addressing a common source of bias that affects a common calibration method relevant to the measurement of trace gases in the atmosphere. This bias arises from the use of regressions in log-linear space without accounting for how the log-transformation impacts the distributions of uncertainty in calculated ion sensitivities. The manuscript is well-written and suitable for publication in AMT following some minor revisions.

Comment 1: It took me a little too long to figure out that in Fig.1 the authors were using a pre-determined fit and sigma to simulate the sensitivities of 100 ions. The default interpretation of these figures is that the plotted line is a fit of the synthetic data, which is exactly backwards in this case. I think a little clarification in the figure caption (and maybe the text) on how the authors simulated the dataset in Fig. 1 would help readers avoid this confusion. Alternatively, Fig. 1 might be more intuitively presented by showing the probability distribution of sensitivities surrounding the assumed fit as shading, rather than through a synthetic dataset plotted as markers (which are too sparse to really see the distribution, unlike e.g. those in Fig S5).

Response: The reviewer is correct that the data are simulated using a pre-determined fit and sigma and we apologize for the lack of clarity in the description. We have added some detailed descriptions on the generation of data on Line 116:

""Sensitivities of 100 simulated analytes are shown Sensitivity and  $dV_{50}$  of 100 analytes are backcalculated from a pre-determined log-linear fit (i.e., slope and intercept of the line) with a distribution of scatter described by  $\sigma_{scatter} = 0.4$  log units (i.e., a factor of 2.5, similar to previously-estimated uncertainty in an iodide-CIMS (Isaacman-Vanwertz et al., 2018))."

We have also revised Figure 1 according to the reviewer's suggestion.

**Figure 1.** Simulated samples relationship between log(sensitivity) and dV50 for an iodide-CIMS based on an assumed log-linear relationship (i.e., slope = -1 log unit/V, maximum sensitivity = 10 @  $dV_{50} \ge 6$ V). Nominal relationship is the black line, with simulated sensitivities for 100 analytes around this relationship as circles. Blue markers demonstrate bias in the average value as described in the text. The shading indicates the probability density of sensitivities around the fitted relationship.

Comment 2: The implicit assumption in this analysis is that the error in sensitivity is normally distributed for all calibrants. Given what we know about the challenges in calibrating sticky molecules, is there a risk that the distribution of errors in a calibration set are not normal, and are actually skewed away from  $\varepsilon$ =0? Since the number of calibrants included in the regression of S vs dV50 is typically very low, it seems like there is significant potential for one or two biased calibrants to shift the S vs dV50 relationship by an amount that is greater than the correction outlined in this manuscript. I would like to stress that this question does not take away from the usefulness of the authors' technique.

Response: The reviewer raises an interesting question about whether the distribution of error may be nonnormal due to issues in real-world sampling. As a note, we interpret here that the reviewer is referring to log-normally distributed error (as opposed to normally distributed), given our discussion of the importance of this aspect. It is an interesting possibility that our sources of error (i.e., sampling line losses) may bias estimates of sensitivity for certain compounds. It is not clear if those errors would be correlated to dV50, so it is not clear that any specific bias would be introduced. Furthermore, because the log-linear fit is usually generated from the data, the error is expected to be log-normally distributed in any case, and any such bias would impact the slope of the nominal relationship. To examine this possibility, we used a Shapiro-Wilk test to invesitgate the log-transformed normality of calibrant sensitivities reported by Mattila et al. (2020) and Iver et al. (2016) and found that their reported sensitivities followed normal distribution after log-transformation. Given the real-world cases, and the empiricism of the fit itself, we believe that a non-normal distribution of calibrant sensitivities would be uncommon (and speculate that biases in the slopes may be more common). However, the mathematical transformation may not work for all cases, and we agree with the reviewer that it is possible the limited number of calibrants for fitting the log-linear relationship may be skewed away from a log-normal distribution if one or two calibrants are significantly biased. In these cases, it might be an indication that the calibration itself is poorly conducted and should be improved by replacing it with other calibrants or using better

instrument operating conditions. We have made some clarifications on the assumption of log-normality on Line 106:

"In such a fit, the error term will be evenly distributed in logarithmic terms is assumed to be normally distributed in logarithmic terms, meaning it is log-normally distributed in linear terms (i.e.,  $\varepsilon$  is normally distributed)."

Comment 3: I am also wondering about the fundamental relationship between binding enthalpy and CIMS sensitivity and whether or not the set of compounds that are easily synthesizable (commercially available) are representative of those that are not. That is to say, the same aspects of chemical structure that make a compound challenging to synthesize (many different functional groups on the same carbon backbone) are also the things that determine how efficiently a molecule can bind to a reagent ion. This is certainly a question beyond the scope of this work, but I think a brief discussion of this weakness in the application of voltage scanning to analytes with unknown structure would fit in well with the parts of this manuscript that discuss what the relationship between S and dV50 means (line 172).

Response: We agree with the reviewer that it is necessary to include some discussions on the weakness of this application. Since the method was originally developed by Iyer et al. (2016), instead of us, we will provide our understanding of the weakness within the context of this work. Iyer et al. (2016) have pointed out the log-linear relationship between I-CIMS sensitivity and binding enthalpy is purely empirical. As excerpted from their work, "the proposed fit to the binding enthalpy is thus purely empirical, with no deeper physical significance beyond the simple fact that the survival probability of an ion-molecule cluster must increase with the strength of its binding." Although they used several liquid standards to validate this empirical relationship, it is possible, as suggested by the reviewer, that the commercially available standards might not be fully representative of those that are not. To answer this question, we have broadened the chemicals from liquid standards to 63 oxidation productions in another work (Bi et al., 2021), currently under review in this same journal, which the reviewer may find interesting. We found that 60% and 80% of the compounds were estimated within a factor of 3 and 10 uncertainties, respectively, indicating that the voltage scan approach has high uncertainties for individual components (Bi et al., 2021). We have added the limitation of the voltage scanning methods on Line 186:

"However, we note that the sensitivity of some compounds predicted by the log-linear relationship between sensitivity and  $dV_{50}$  may have high uncertainty, likely due to the empirical nature of the relationship (Bi et al., 2021)."

Comment 4: 58: There is a step missing between what is shown in Lopez-Hilfiker et al 2016 (dV50 is an indicator of binding enthalpy) and the authors' conclusion that sensitivity and dV50 have a log-linear relationship. I think an additional citation to Iyer et al. 2016 is warranted here.

Response: We would like to thank the reviewer for pointing out the missing citation. The additional citation has been added on Line 58

"The parameter,  $dV_{50}$ , which is the voltage difference where signals of a compound are at half-maximum, is reported to be an indicator of the binding-enthalpy of the adduct (Iyer et al., 2016; Lopez-Hilfiker et al., 2016)."

Comment 5: 115: The phrase "analyte-reagent ion binding-enthalpy" was confusing to read because of all the hyphens. I would reword this to "the binding enthalpy of the analyte with the reagent ion" or similar.

Response: As suggested by the reviewer, we have revised the manuscript on Line 116 to improve the clarity of the text.

"analyte-reagent ion binding-enthalpy the binding enthalpy of the analyte with the reagent ion"

Comment 6: 224: Framing slope as a negative value has flipped the reader around from what is plotted in Fig. 4. I follow the authors' reasoning and the math, and I think that readers would follow along more easily if the authors did some combination of 1) including  $\Delta dV50$  as a second x-axis in Fig. 4.2) explicitly address at ~line 224 that the use of  $\Delta dV50$  has changed the sign of the slope compared to when it is plotted against dV50

Response: We apologize for the lack of clarity on the negative value of the slope. We have revised Figure 4 and the manuscript based on the two suggestions from the reviewer. Some additional discussions are added on Line 230:

"The use of  $\Delta dV_{50}$  has changed the sign of the slope compared to when it is plotted against  $dV_{50}$  (top x-axis in Figure 4). The slope is defined as change in log(sensitivity) per unit  $\Delta dV_{50}$ , and is therefore necessarily a negative value (i.e., sensitivity decreases with  $\Delta dV_{50}$ )."

Figure 4. Nominal (black line) log-linear relationship between sensitivity and  $dV_{50}$  following a typical iodide-CIMS calibration form, with labels on the four sources of uncertainty described in the text. Each circle represents one ion with the distribution of uncertainties listed. Dashed line illustrates one possible relationship representing one standard deviation away from each nominal value. Note that the use of  $\Delta dV_{50}$  in Equation 9 has changed the sign of the slope compared to when it is plotted against  $dV_{50}$ .

Comment 7: 330: Just I-CIMS calibrated through voltage scanning, not I-CIMS in general.

Response: We agree with the reviewer and have revised the manuscript on Line 350 based on the reviewer's comment.

"However, when a log transformation is used to linearize the relationship between sensitivity and a parameter, as in the case of an iodide-CIMS an iodide-CIMS is calibrated by the voltage scanning method utilizing the linear relationship between log transformation of sensitivity and a parameter, an inherent bias is introduced into the sensitivity of analytes."

Comment 8: Fig. S5: This figure is helpful for seeing the interplay between the shape of the voltage scanning dataset and the bias introduced at different dV50. I found myself wanting to know the average % bias in the measured mass of analytes for the No-Correction fit in each case. It's another number on a very full figure, but I think it'd be worth it.

Response: We would like to thank the reviewer for improving the quality of Figure S5. The average % bias in total mass of analytes has been added to Figure S5 and the updated Figure S5 is included in the response below:

---

## Author Comment (AC2)

The authors would like to thank both reviewers for the feedback on the manuscript. Those comments are insightful, and the suggestions have helped improve the quality of the manuscript substantially. We have made revisions to the manuscript according to the reviewers' comments and the extra experimental findings. The colorings of text in the reviewer response are:

- Light blue: Original reviewer comments
- Dark blue: Text added in the revision while  are the text deleted in the revised manuscript.
- Black: Original text in the submitted version of the manuscript and authors' response to the comments and others.

Note that the line number in the response is based on the revised clean-version manuscript.

Reviewer 1:

This manuscript presents a method for addressing a common source of bias that affects a common calibration method relevant to the measurement of trace gases in the atmosphere. This bias arises from the use of regressions in log-linear space without accounting for how the log-transformation impacts the distributions of uncertainty in calculated ion sensitivities. The manuscript is well-written and suitable for publication in AMT following some minor revisions.

Comment 1: It took me a little too long to figure out that in Fig.1 the authors were using a pre-determined fit and sigma to simulate the sensitivities of 100 ions. The default interpretation of these figures is that the plotted line is a fit of the synthetic data, which is exactly backwards in this case. I think a little clarification in the figure caption (and maybe the text) on how the authors simulated the dataset in Fig. 1 would help readers avoid this confusion. Alternatively, Fig. 1 might be more intuitively presented by showing the probability distribution of sensitivities surrounding the assumed fit as shading, rather than through a synthetic dataset plotted as markers (which are too sparse to really see the distribution, unlike e.g. those in Fig S5).

Response: The reviewer is correct that the data are simulated using a pre-determined fit and sigma and we apologize for the lack of clarity in the description. We have added some detailed descriptions on the generation of data on Line 116:

"" Sensitivity and $dV_{50}$ of 100 analytes are back-calculated from a pre-determined log-linear fit (i.e., slope and intercept of the line) with a distribution of scatter described by $\sigma_{scatter} = 0.4$ log units (i.e., a factor of 2.5, similar to previously-estimated uncertainty in an iodide-CIMS (Isaacman-Vanwertz et al., 2018))."

We have also revised Figure 1 according to the reviewer's suggestion.

[Figure]

**Figure 1.** Simulated samples  between log(sensitivity) and dV50 for an iodide-CIMS based on an assumed log-linear relationship (i.e., slope = -1 log unit/V, maximum sensitivity = 10 @ $dV_{50} \geq 6$ V). Nominal relationship is the black line, with simulated sensitivities for 100 analytes around this relationship as circles. Blue markers demonstrate bias in the average value as described in the text. The shading indicates the probability density of sensitivities around the fitted relationship.

Comment 2: The implicit assumption in this analysis is that the error in sensitivity is normally distributed for all calibrants. Given what we know about the challenges in calibrating sticky molecules, is there a risk that the distribution of errors in a calibration set are not normal, and are actually skewed away from ε=0? Since the number of calibrants included in the regression of S vs dV50 is typically very low, it seems like there is significant potential for one or two biased calibrants to shift the S vs dV50 relationship by an amount that is greater than the correction outlined in this manuscript. I would like to stress that this question does not take away from the usefulness of the authors' technique.

Response: The reviewer raises an interesting question about whether the distribution of error may be non-normal due to issues in real-world sampling. As a note, we interpret here that the reviewer is referring to log-normally distributed error (as opposed to normally distributed), given our discussion of the importance of this aspect. It is an interesting possibility that our sources of error (i.e., sampling line losses) may bias estimates of sensitivity for certain compounds. It is not clear if those errors would be correlated to dV50, so it is not clear that any specific bias would be introduced. Furthermore, because the log-linear fit is usually generated from the data, the error is expected to be log-normally distributed in any case, and any such bias would impact the slope of the nominal relationship. To examine this possibility, we used a Shapiro-Wilk test to invesitgate the log-transformed normality of calibrant sensitivities reported by Mattila et al. (2020) and Iyer et al. (2016) and found that their reported sensitivities followed normal distribution after log-transformation. Given the real-world cases, and the empiricism of the fit itself, we believe that a non-normal distribution of calibrant sensitivities would be uncommon (and speculate that biases in the slopes may be more common). However, the mathematical transformation may not work for all cases, and we agree with the reviewer that it is possible the limited number of calibrants for fitting the log-linear relationship may be skewed away from a log-normal distribution if one or two calibrants are significantly biased. In these cases, it might be an indication that the calibration itself is poorly conducted and should be improved by replacing it with other calibrants or using better

instrument operating conditions. We have made some clarifications on the assumption of log-normality on Line 106:

"In such a fit, the error term is assumed to be normally distributed in logarithmic terms, meaning it is log-normally distributed in linear terms (i.e., $\varepsilon$ is normally distributed)."

Comment 3: I am also wondering about the fundamental relationship between binding enthalpy and CIMS sensitivity and whether or not the set of compounds that are easily synthesizable (commercially available) are representative of those that are not. That is to say, the same aspects of chemical structure that make a compound challenging to synthesize (many different functional groups on the same carbon backbone) are also the things that determine how efficiently a molecule can bind to a reagent ion. This is certainly a question beyond the scope of this work, but I think a brief discussion of this weakness in the application of voltage scanning to analytes with unknown structure would fit in well with the parts of this manuscript that discuss what the relationship between S and dV50 means (line 172).

Response: We agree with the reviewer that it is necessary to include some discussions on the weakness of this application. Since the method was originally developed by Iyer et al. (2016), instead of us, we will provide our understanding of the weakness within the context of this work. Iyer et al. (2016) have pointed out the log-linear relationship between I-CIMS sensitivity and binding enthalpy is purely empirical. As excerpted from their work, "the proposed fit to the binding enthalpy is thus purely empirical, with no deeper physical significance beyond the simple fact that the survival probability of an ion−molecule cluster must increase with the strength of its binding." Although they used several liquid standards to validate this empirical relationship, it is possible, as suggested by the reviewer, that the commercially available standards might not be fully representative of those that are not. To answer this question, we have broadened the chemicals from liquid standards to 63 oxidation productions in another work (Bi et al., 2021), currently under review in this same journal, which the reviewer may find interesting. We found that 60% and 80% of the compounds were estimated within a factor of 3 and 10 uncertainties, respectively, indicating that the voltage scan approach has high uncertainties for individual components (Bi et al., 2021). We have added the limitation of the voltage scanning methods on Line 186:

"However, we note that the sensitivity of some compounds predicted by the log-linear relationship between sensitivity and $dV_{50}$ may have high uncertainty, likely due to the empirical nature of the relationship (Bi et al., 2021)."

Comment 4: 58: There is a step missing between what is shown in Lopez-Hilfiker et al 2016 (dV50 is an indicator of binding enthalpy) and the authors' conclusion that sensitivity and dV50 have a log-linear relationship. I think an additional citation to Iyer et al. 2016 is warranted here.

Response: We would like to thank the reviewer for pointing out the missing citation. The additional citation has been added on Line 58

"The parameter, $dV_{50}$, which is the voltage difference where signals of a compound are at half-maximum, is reported to be an indicator of the binding-enthalpy of the adduct (Iyer et al., 2016; Lopez-Hilfiker et al., 2016)."

Comment 5: 115: The phrase "analyte-reagent ion binding-enthalpy" was confusing to read because of all the hyphens. I would reword this to "the binding enthalpy of the analyte with the reagent ion" or similar.

Response: As suggested by the reviewer, we have revised the manuscript on Line 116 to improve the clarity of the text.

" the binding enthalpy of the analyte with the reagent ion"

Comment 6: 224: Framing slope as a negative value has flipped the reader around from what is plotted in Fig. 4. I follow the authors' reasoning and the math, and I think that readers would follow along more easily if the authors did some combination of 1) including $\Delta dV50$ as a second x-axis in Fig. 4 2) explicitly address at ~line 224 that the use of $\Delta dV50$ has changed the sign of the slope compared to when it is plotted against $dV50$

Response: We apologize for the lack of clarity on the negative value of the slope. We have revised Figure 4 and the manuscript based on the two suggestions from the reviewer. Some additional discussions are added on Line 230:

"The use of $\Delta dV_{50}$ has changed the sign of the slope compared to when it is plotted against $dV_{50}$ (top x-axis in Figure 4). The slope is defined as change in log(sensitivity) per unit $\Delta dV_{50}$, and is therefore necessarily a negative value (i.e., sensitivity decreases with $\Delta dV_{50}$)."

[Figure]

Figure 4. Nominal (black line) log-linear relationship between sensitivity and $dV_{50}$ following a typical iodide-CIMS calibration form, with labels on the four sources of uncertainty described in the text. Each circle represents one ion with the distribution of uncertainties listed. Dashed line illustrates one possible relationship representing one standard deviation away from each nominal value. Note that the use of $\Delta dV_{50}$ in Equation 9 has changed the sign of the slope compared to when it is plotted against $dV_{50}$.

Comment 7: 330: Just I-CIMS calibrated through voltage scanning, not I-CIMS in general.

Response: We agree with the reviewer and have revised the manuscript on Line 350 based on the reviewer's comment.

"However, when  an iodide-CIMS is calibrated by the voltage scanning method utilizing the linear relationship between log transformation of sensitivity and a parameter, an inherent bias is introduced into the sensitivity of analytes."

Comment 8: Fig. S5: This figure is helpful for seeing the interplay between the shape of the voltage scanning dataset and the bias introduced at different dV50. I found myself wanting to know the average % bias in the measured mass of analytes for the No-Correction fit in each case. It's another number on a very full figure, but I think it'd be worth it.

Response: We would like to thank the reviewer for improving the quality of Figure S5. The average % bias in total mass of analytes has been added to Figure S5 and the updated Figure S5 is included in the response below:

[Figure]

**Figure S5.** The influence of $\sigma_{scatter}$ (log units) and $\sigma_{slope}$ (log units/volt) of the analyte sensitivity distribution on the fitted. relationship with parameter-explicit bias corrections and simplified effective $\sigma_{scatter}$ corrections (Eq. 11). The average bias in total mass of analytes without bias correction is displayed at the bottom of the figure. Each circle represents one ion with the distribution of $\sigma_{scatter}$ and $\sigma_{slope}$ uncertainties listed. The number of simulated ions, $\sigma_{dV50, max}$, and $\sigma_{Smax}$ in the dataset are 10000, 0 V, and 0%, respectively.

Reviewer 2

In this paper, the authors examined the instrument bias in a log-linear based calibration curve and provide a method to correct that bias, which are useful for researchers in atmospheric chemistry and sciences. I think this work is suitable for AMT journal topic. I would recommend accepting this paper with a couple of revision and re-consideration mentioned below.

**Specific comments:**

Comment 1: 1. In Figures 1, 4, and 7, the authors used 100, 1000, and 10000 ions/analytes for the simulation of sample relationship between sensitivity and dV50. I have a couple of questions here: (i) What are those ions/analytes in real? (Are those able to exist in real?) (ii) How were the sensitivities and dV50 for individual ions/analytes determined? It would be good if more detailed explanations regarding those matters are added in the caption or the main text.

Response: We apologize for the lack of clarity in the method simulating the dataset in the figures. In all three cases, those ions are simulated potential analytes having properties prescribed based on some pre-defined nominal fit and uncertainties.The simulated ions are generated based on a pre-defined slope, and/or maximum sensitivity in the log-linear calibration relationship used by Isaacman-Vanwertz et al. (2018) while the uncertainties ($\sigma_{scatter}$, $\sigma_{slope}$, $\sigma_{dV50,max}$) are either directly obtained from Isaacman-Vanwertz et al. (2018) or assigned in this study to probe the influence of each parameter on the potential bias. Each ion is assigned an arbitrary dV50, from which a sensitivity is generated based on the distribution of defined uncertainty from the defined log-linear fit . Therefore, those ions are not real since they are simulated dataset. However, since the log-linear calibration relationship reported by Isaacman-Vanwertz et al. (2018) was developed based on experimental chamber data, the simulated dataset in this study should still represent the real-world scenario in some cases.

To improve the clarity of the data simulation method, we have made several revisions in the manuscript as well as in the caption of the figures:

1) Revisions on Line 116

   " Sensitivity and $dV_{50}$ of 100 analytes are back-calculated from a pre-defined log-linear fit (i.e., slope and intercept of the line) with a distribution of scatter described by $\sigma_{scatter} = 0.4$ log units (i.e., a factor of 2.5, similar to previously-estimated uncertainty in an iodide-CIMS (Isaacman-Vanwertz et al., 2018))."

2) Revisions of figure captions

   Figure 1. Simulated samples  between log(sensitivity) and dV50 for an iodide-CIMS based on an assumed log-linear relationship (i.e., slope = -1 log unit/V, maximum sensitivity = 10 @ $dV_{50} \geq 6$ V). Nominal relationship is the black line, with simulated sensitivities for 100 analytes around this relationship as circles. Blue markers demonstrate bias in the average value as described in the text. The shading indicates the probability density of sensitivities around the fitted relationship.

   Figure 4. Nominal (black line) log-linear relationship between sensitivity and $dV_{50}$ following a typical iodide-CIMS calibration form, with labels on the four sources of uncertainty described in the text. Each circle represents one

simulated ion generated from the log-linear relationship determined by Isaacman-Vanwertz et al. (2018) within a distribution of uncertainty defined by the values listed in the figure. Dashed line illustrates one possible relationship representing one standard deviation away from each nominal value.

Figure 7. The nominal (black line), parameter-explicit bias corrected (blue line), and $\sigma^{eff}_{scatter}$ bias corrected (green line) log-linear relationship between sensitivity and $dV_{50}$.  Each pink circle represents one simulated ion generated from the log-linear relationship determined by Isaacman-Vanwertz et al. (2018) with uncertainties replaced with those listed in the figure.

Comment 2: 2. Also I think, it would be good if the author can describe how the bias correction method can correct the measured results in real. According to previous literature (e.g. Iyer et al., JPCA 2016; Lopez-Hilfiker et al., AMT 2016), the sensitivities and dV50 of several compounds for iodide CIMS have been measured. Using those values, the usefulness of the method developed in this work should be able to be discussed.

Response: We have included some discussions for the implementation of this technique in real-world cases on Line 327. However, it seems to us that our discussion is not clear enough to provide a full answer to the reviewer's question. In addition to reporting the calculated $\sigma^{eff}_{scatter}$ of the two previous studies, we have added estimations of the average % bias in summed mass without bias correction to show the strength of this correction technique in real-world cases. To be clear, bias will not influence measurements in cases for which a calibrated sensitivity is used, only in cases in which voltage scanning is used to estimate sensitivity. We have revised this paragraph slightly to stress the usefulness of the method as suggested by the reviewer on Line 327 and clarify that this applies only to cases where voltage scanning is used to determine sensitivity:

"In two previous studies implementing the voltage scan calibration, we found that the calculated $\sigma^{eff}_{scatter}$ can be as low as 0.012 (Mattila et al., 2020), or as high as 0.29 (Lopez-Hilfiker et al., 2016; Iyer et al., 2016). Based on the log-linear fit and the calculated $\sigma^{eff}_{scatter}$ in the two previous studies, the average bias in the summed mass of 100 simulated ions would be approximate 2% and 28% in Mattila et al. (2020) and Lopez-Hilfiker et al. (2016), respectively, if sensitivities were determined by voltage scanning. However, these two cases represent voltage scanning of two different instrument voltage regions and are calculated using limited number of calibrants. The potential range of the $\sigma^{eff}_{scatter}$ therefore remains unclear and future work is needed to examine this approach in real-world applications, and the real potential for bias in voltage scanning approaches."

Comment 3: 3. Sensitivities in iodide-CIMS should have uncertainties arising from biding energy between I⁻ and analyte. I am wondering how large or small these uncertainties are compared to the bias arising from instrumental parameters. It is possible to quantitatively discuss it?

Response: The reviewer raises an interesting and important question. It is important to first clarify that the bias we discuss here is distinct from uncertainty in the calibration. The uncertainty is the estimation of error and will have some distribution, the broadness of which is an indication of precision, around some central tendency, which has some accuracy. The bias is a systematic error in the accuracy. The technique demonstrated in this study is to correct bias and improve the accuracy of the voltage scanning method while the instrumental-related parameters are mostly associated with the precision-related uncertainty. We agree that these issues are not wholly independent; a small bias is probably not a major concern if the instrument has very poor precision (for instance, the case shown in the green line of Figure 3).

In a separate study, we have quantified the uncertainties of the voltage scanning method (Bi et al., 2021). We found that 60% and 80% of 63 oxidation products are estimated within a factor of 3 and 10 uncertainties, respectively, indicating that the voltage scanning approach has high uncertainties for individual compounds. In contrast to the high individual uncertainty, the summed mass of analytes in an oxidation experiment can be estimated within 30% by the voltage scanning approach (Bi et al., 2021). The sample real-world data used in this work from Isaacman-VanWertz et al. (2018) found a bias of roughly 30%, suggesting that the bias is similar in magnitude to precision error and thus probably relevant. However, since the bias may vary significantly depending on the uncertainties of the log-linear fit (Figure 5) and the sensitivity of a specific analyte or a group of analytes (Figure S5), the bias can be either larger or smaller than the instrument-related uncertainties.

For example, as shown in Figure S5g, a less sensitive compound with a $dV_{50} = 4.0$ V should have corrected sensitivity one order of magnitude higher than the uncorrected one, which is comparable or larger than the measurement uncertainty (a factor of $3 - 10$ as reported by Bi et al. (2021)). In contrast, a very sensitive compound with a $dV_{50} > 6.0$ V, may have corrected sensitivity similar to the uncorrected one. The bias of highly sensitive compounds is substantially lower than the measurement uncertainty. It would also be interesting to compare the bias to the measurement uncertainty in summed mass of analytes in an oxidation experiment. As we added to Figure S5, the bias in the summed mass can be as low as 8%, which is similar to measured uncertainty, and as high as 300%, which is an order of magnitude higher than the measurement uncertainty in summed mass (30% as reported by Bi et al. (2021)).

In summary, we believe that the relative difference in mathematical bias versus measured uncertainty can vary significantly depending on the sensitivity of a specific analyte or a group of analytes and instrumental operating conditions. However, regardless of the precision, the accuracy of the data would in all cases be improved by a parameter-explicit bias correction. We have added some discussions of this comparison in the manuscript on Line 337 and revised Figure S5:

"Uncertainty of instrumental measurements is frequently reported in the literature, which is typically a measure of the combined instrument precision and accuracy. In contrast, bias represents a systematic error in the accuracy and is distinct from these reported uncertainties. It is theoretically worth comparing the relative magnitude of the two types of errors, as a small bias would likely be negligible in the case of large uncertainty. However, this is difficult as bias may vary significantly depending on the uncertainties of the log-linear fit, with examples shown in Figure S5 ranging from 8 to 300% bias in summed mass of analytes. Recent work by Bi et al. (2021) found uncertainties of a factor of 3-10 for individual ions and ~30% for the sum of many ions using the voltage scanning method. This summed uncertainty is comparable in scale to the bias determined for the data from Isaacman-VanWertz et al. (2018), indicating bias is likely non-negligible. For individual ions, the importance of bias correction depends strongly on the $dV_{50}$ of the compound and the scale of the bias correction, though a parameter-explicit bias correction always increases accuracy."

[Figure]

**Figure S5.** The influence of $\sigma_{scatter}$ (log units) and $\sigma_{slope}$ (log units/volt) of the analyte sensitivity distribution on the fitted. relationship with parameter-explicit bias corrections and simplified effective $\sigma_{scatter}$ corrections (Eq. 11). The average bias in total mass of analytes without bias correction is displayed at the bottom of the figure. Each circle represents one ion with the distribution of $\sigma_{scatter}$ and $\sigma_{slope}$ uncertainties listed. The number of simulated ions, $\sigma_{dV50, max}$, and $\sigma_{Smax}$ in the dataset are 10000, 0 V, and 0%, respectively.

Comment 4: 4. In Introduction section, the authors mentioned this work focuses on the calibrations of analytes in an iodide-CIMS. So, maybe it would be better the title also says "iodide chemical ionization mass spectrometry".

Response: We appreciate the reviewer's suggestion on the potential change of the title. Although this bias correction technique is demonstrated using iodide-CIMS as an example, we believe it can be applied to any cases implementing similar voltage scanning method. In fact, this bias demonstrated in this study exists mathematically in any log-linear relationship between predicted and measured parameters. The correction technique should be able to be used even outside of the field of mass spectrometry. Therefore, we decide not to revise the title of this manuscript. We have already included some discussions on the potential wider application of this correction technique on Line 354:

"Although the discussions of this work mainly focus on iodide-CIMS, we believe that this correction can be applied to other CIMS or more broadly, other atmospheric measurement instruments using log-linear calibration relationships."

References:

Bi, C., Krechmer, J. E., Frazier, G. O., Xu, W., Lambe, A. T., Claflin, M. S., Lerner, B. M., Jayne, J. T., Worsnop, D. R., Canagaratna, M. R., and Isaacman-VanWertz, G.: Quantification of Isomer-Resolved Iodide CIMS Sensitivity and Uncertainty Using a Voltage Scanning Approach, Atmos. Meas. Tech. Discuss., 2021, 1-27, 10.5194/amt-2021-164, 2021.
Isaacman-Vanwertz, G., Massoli, P., O'Brien, R., Lim, C., Franklin, J. P., Moss, J. A., Hunter, J. F., Nowak, J. B., Canagaratna, M. R., Misztal, P. K., Arata, C., Roscioli, J. R., Herndon, S. T., Onasch, T. B., Lambe, A. T., Jayne, J. T., Su, L., Knopf, D. A., Goldstein, A. H., Worsnop, D. R., and Kroll, J. H.: Chemical evolution of atmospheric organic carbon over multiple generations of oxidation, Nature Chemistry, 10, 462-468, 10.1038/s41557-018-0002-2, 2018.
Iyer, S., Lopez-Hilfiker, F., Lee, B. H., Thornton, J. A., and Kurtén, T.: Modeling the Detection of Organic and Inorganic Compounds Using Iodide-Based Chemical Ionization, Journal of Physical Chemistry A, 120, 576-587, 10.1021/acs.jpca.5b09837, 2016.
Lopez-Hilfiker, F. D., Iyer, S., Mohr, C., Lee, B. H., D'Ambro, E. L., Kurtén, T., and Thornton, J. A.: Constraining the sensitivity of iodide adduct chemical ionization mass spectrometry to multifunctional organic molecules using the collision limit and thermodynamic stability of iodide ion adducts, Atmospheric Measurement Techniques, 9, 1505-1512, 10.5194/amt-9-1505-2016, 2016.
Mattila, J. M., Lakey, P. S. J., Shiraiwa, M., Wang, C., Abbatt, J. P. D., Arata, C., Goldstein, A. H., Ampollini, L., Katz, E. F., Decarlo, P. F., Zhou, S., Kahan, T. F., Cardoso-Saldaña, F. J., Ruiz, L. H., Abeleira, A., Boedicker, E. K., Vance, M. E., and Farmer, D. K.: Multiphase Chemistry Controls Inorganic Chlorinated and Nitrogenated Compounds in Indoor Air during Bleach Cleaning, Environmental Science and Technology, 54, 1730-1739, 10.1021/acs.est.9b05767, 2020.